# Investigation of HIV-1 Gag binding with RNAs and lipids using Atomic Force Microscopy

**Shaolong Chen[1], Jun Xu[1], Mingyue Liu[1], A. L. N. Rao[2], Roya Zandi[1], Sarjeet S. Gill[3], Umar Mohideen[1]***

**1** Department of Physics & Astronomy, University of California, Riverside, California, United States of America, **2** Department of Plant Pathology & Microbiology, University of California, Riverside, California, United States of America, **3** Department of Cell Biology & Neuroscience, University of California, Riverside, California, United States of America

\* umar.mohideen@ucr.edu

**Data Availability Statement:** All relevant data are within the manuscript and its Supporting Information files.

**Funding:** The author(s) received no specific funding for this work.

## Abstract

Atomic Force Microscopy was utilized to study the morphology of Gag, ΨRNA, and their binding complexes with lipids in a solution environment with 0.1Å vertical and 1nm lateral resolution. TARpolyA RNA was used as a RNA control. The lipid used was phospha-tidylino-sitol-(4,5)-bisphosphate (PI(4,5)P2). The morphology of specific complexes Gag-ΨRNA, Gag-TARpolyA RNA, Gag-PI(4,5)P2 and PI(4,5)P2-ΨRNA-Gag were studied. They were imaged on either positively or negatively charged mica substrates depending on the net charges carried. Gag and its complexes consist of monomers, dimers and tetramers, which was confirmed by gel electrophoresis. The addition of specific ΨRNA to Gag is found to increase Gag multimerization. Non-specific TARpolyA RNA was found not to lead to an increase in Gag multimerization. The addition PI(4,5)P2 to Gag increases Gag multimeriza-tion, but to a lesser extent than ΨRNA. When both ΨRNA and PI(4,5)P2 are present Gag undergoes comformational changes and an even higher degree of multimerization.

## Introduction

Human immunodeficiency virus (HIV) is a retrovirus with a diploid genome of single-stranded RNA [1–4]. Two types of HIV, HIV-1 and HIV-2, have been reported [5]. HIV is a spherical membrane-bound virus with a diameter ranging from 100 nm to 150 nm. The formation of infectious HIV is considered to have three stages: (1) assembly, (2) budding and release, and (3) maturation [1, 6–8]. In the first assembly stage the components are encapsulated to create the immature virion at the plasma membrane. In the second budding and release stage, the immature virion forms its lipid envelope and buds from the plasma membrane. The last maturation stage is where the immature virion undergoes conformational changes to form the mature infectious virus. In all these three stages, the most important structural component of HIV is the genetic polyprotein precursor Gag. The HIV immature virion has about 2500–5000 copies of the Gag polyprotein [1, 3]. The approximate mass of Gag is 55kDa [8]. Extended Gag is thought to have a cylindrical shape with a length of 20-30nm and a diameter of 2-3nm [2]. From N-terminus to C-terminus, Gag consists of six structural

**Competing interests:** The authors have declared that no competing interests exist.

domains: a matrix (MA) domain with amino-terminal myristylation (Myr), a capsid (CA) domain, spacer peptides 1 (SP1), a nucleocapsid (NC) domain, spacer peptides 2 (SP2), and a p6 domain, respectively [1, 6,7, 9]. The three primary functional domains of the Gag are MA, CA and NC. The HIV Gag precursor in the immature virion is radially oriented [1]. The N-terminal MA domain is bound to the inner leaflet of the lipid membrane. The C-terminal NC domain binds with two copies of viral genomic RNA. The CA domain interacts with each other forming a hexagonal lattice. In the final stages, the HIV virion becomes mature using a viral protease that cleaves the Gag in a specific order as discussed in Ref. [10].

The MA domain of HIV-1 Gag with 104 amino acids exhibits five alpha helices and a triple-stranded beta sheet [11–13]. MA has a myristoylated fatty acid group at its N-terminus which is responsible for Gag assembly and targets to the phospha-tidylinositol-(4,5)-bisphosphate (PI(4,5)P2) on the lipid membrane [14,15]. The 14-carbon myristoylated group is initially sequestered in a hydrophobic cleft of the MA domain, but is later exposed and facilitates Gag binding with the lipid membrane. The exposure of the myristyl acid group is activated by PI(4,5)P2 which is abundant in the plasma membrane. The CA domain which is responsible for Gag-Gag interaction is separated into two parts, the N-terminal domain (CA$^{NTD}$) and the C-terminal domain (CA$^{CTD}$) connected by a flexible linker. The arrowhead-like shaped CA$^{NTD}$ containing seven alpha helices is essential for the formation of a conical outer shell of the capsid core [16, 17]. The CA$^{NTD}$ forms hexameric rings with an approximate spacing of around 8 nm as observed with cryo-electron microscopy (cET) [18, 19]. The CA domain plays a crucial role in the formation of both immature and mature virions. In the mature virus, the mature capsid core consists of 1000–1500 copies of the CA protein assembled into a hexameric lattice with a spacing of 10 nm rather than 8nm which is the spacing of CA hexamers in the immature virions [3]. The NC domain is critical for the genomic viral RNA recognition, interaction and dimerization. The NC domain binds Gag to the RNA genome through nonspecific interaction as well as specific binding to the stem-loop 3 (SL3) in the packaging signal Ψ (ΨRNA) [20]. In the inner core of the mature virus, the viral genomic RNA is wrapped around 1500–2000 copies of NC proteins [21]. Within the NC domain, there are two CCHC type zinc fingers, which are crucial for specific ΨRNA binding and genomic viral RNA packaging [22, 23, 24].

HIV-1 genome is a RNA sequence that has 9173 nucleotides [9, 25]. It is central to many steps of the replication process, such as transcription, genomic dimerization, , HIV genome packagingetc. At the 5' untranslated region (UTR) of HIV-1 genomic RNA, there are many critical regions which are thought necessary for genome dimerization and binding with Gag: the transactivation response stem-loop (TAR), the polyadenylation stem-loop (polyA), the prime binding site (PBS) and the packaging signal domain Ψ. Of the 104 nucleotides of HIV viral RNA, the first 1–57 nucleotides is TAR with 58–104 nucleotides being polyA [26, 27]. The mass of the TARpolyA RNA is around 34kDa. The TARpolyA RNA sequence used in the experiments is shown in Fig 1A [22]. It plays an essential role in HIV genome packaging and reverse transcription [26]. In addition to the dimerization initiation site (DIS) in Ψ, the TAR may also facilitate HIV genomic RNA dimerization when the NC protein is present to form TAR-TAR dimers [28]. The packaging signal Ψ contains about 109 nucleotides [26, 29]. The mass of the ΨRNA is about 36kDa. The ΨRNA sequence used in our work is shown in Fig 1B where most of nucleotides are paired with each other [26]. It contains 3 stem loops SL1, SL2, and SL3. SAXS studies show that ΨRNA adopts an unfolded conformation where all stem loops are open for later interaction with both viral and host elements. The ΨRNA binds with the NC protein and is crucial for packaging of HIV genomic RNA. Within ΨRNA, SL1 contains a palindromic sequence DIS that is responsible for HIV genomic RNA dimerization and Gag binding [30–32]. SL2 includes the splice donor site (SD) that is used to produce spliced

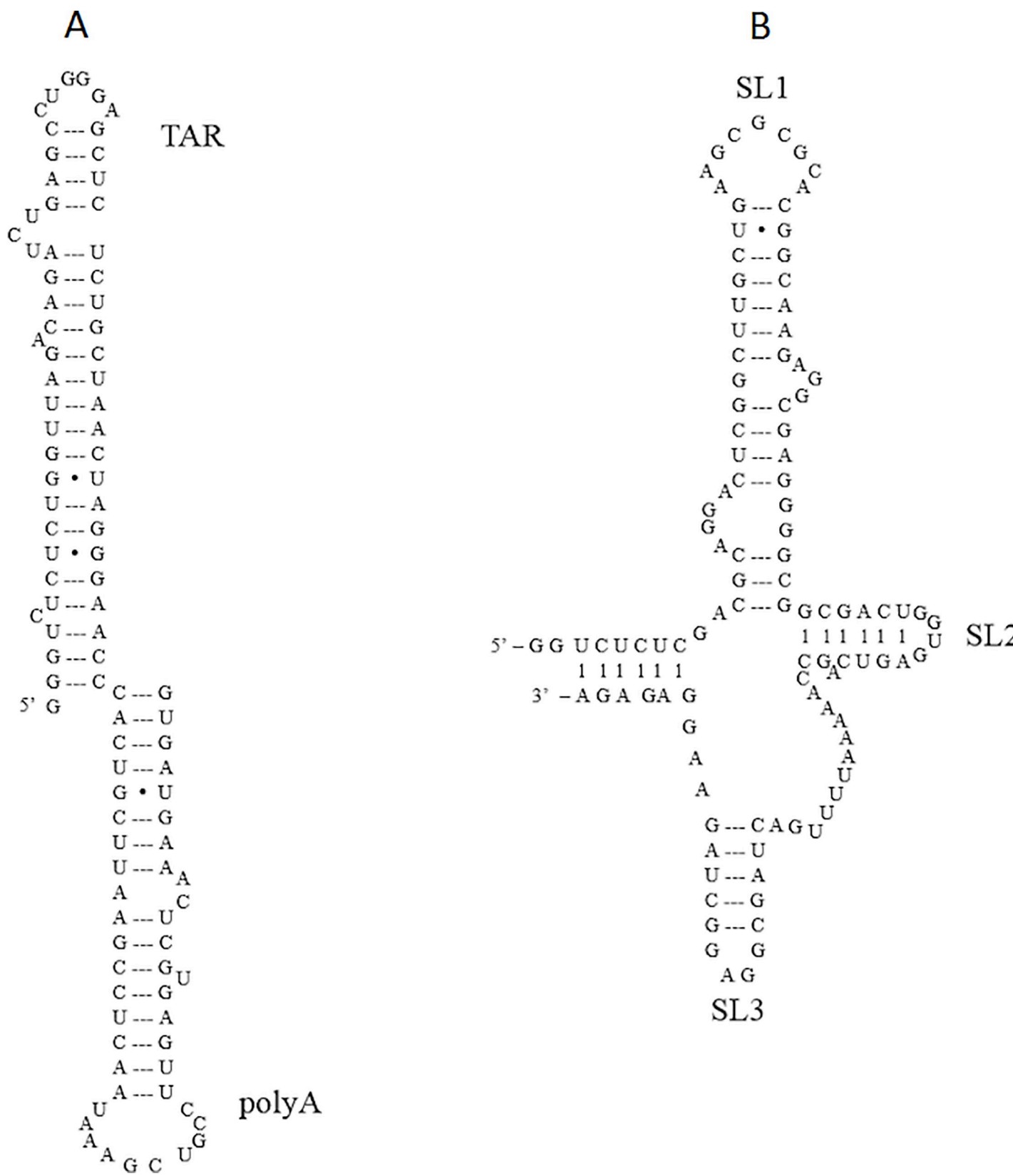

**Fig 1. RNA sequences used.** HIV-1 genomic RNA partial sequence constructs used in this work. (A) TARpolyA RNA. (B) ΨRNA.

message RNAs. SL3 is required for both viral RNA dimerization and packaging [33]. In addition to SL1, SL3 is also a high affinity Gag binding site [34].

PI(4,5)P2 belongs to the negatively charged lipid family known as phosphoinositide [35]. It consists of a glycerol backbone with one saturated fatty acid chain at position 1' and one unsaturated fatty acid chain at position 2' and a phosphoinositol headgroup at position 3'. PI(4,5)P2 serves as a raft for HIV-1 Gag targeting to the plasma membrane and thus regulates the HIV assembly [35, 36]. In particular, PI(4,5)P2 binds with the myristylated MA domain of Gag through a highly specific interaction. First, the phosphoinositol headgroup and the 2' unsaturated fatty acid chain of PI(4,5)P2 insert into a hydrophobic pocket in the MA domain. This then triggers the exposure of the myristylated group for insertion into the lipid bilayer membrane [37–40]. Previous research has suggested that Gag has distinct binding modes with different RNAs [22]. For non-specific TARpolyA RNA, both the NC and MA domains are bound to TARpolyA RNA. While in the case of specific ΨRNA, only the NC domain was found to bind with ΨRNA [22]. In the latter case, MA domain is left free for later interaction with lipid PI(4,5)P2. The motivation of this work is to utilize a high-resolution technique such as Atomic Force Microscopy (AFM) to explore HIV-1 Gag binding with different RNAs and lipids. First, we studied the morphology of individual components of Gag, ΨRNA, and TARpolyA RNA separately as controls. Next, to see the effect of the addition of specific and non-specific RNAs, we investigated the effect on the Gag complex formation by adding either ΨRNA or TARpolyA RNA to Gag. Then, the effect of adding lipid PI(4,5)P2 into Gag was also studied. Finally, the influence of the addition of both ΨRNA and PI(4,5)P2 was examined to understand their collective effect on Gag.

## Materials and methods

### Materials

HIV-1 Gag used in our experiments was obtained from Dr. Alan Rein and Dr. S.A.K. Datta. It lacks the myristylated group and p6 and thus usually is referred to as GagΔP6 [22]. Both ΨRNA and TARpolyA RNA were obtained from Dr. Karin Musier-Forsyth and Dr. E.D. Olson [36]. Brain PI(4,5)P2 (L-α-phosphatidylinositol-4,5-bisphosphate) was purchased from Avanti Polar Lipids (Alabaster, AL, US).

### Preparation of samples

HIV-1 GagΔP6 was originally at 30μM in the buffer containing 20 mM Tris-HCl (pH 7.5), 0.5 M NaCl, 10% (v/v) glycerol, 5 mM DTT (dithiothreitol). It was diluted to 0.5μM before AFM imaging with HEPES buffer, which contained 20mM, HEPES (pH 7.5), 1mM MgCl$_2$, 50mM NaCl, 10μM TCEP (tris-2-carboxy-ethyl phosphine) 5 mM βME (β-mercaptoethanol). ΨRNA and TARpolyA RNA were originally in the same HEPES buffer at a concentration of 74.18μM and 119μM, respectively. Both RNAs need to be refolded before use. The protocol followed for refolding RNA was as follows. First, 22.2μL 74.18μM ΨRNA (or 13.8uL 119μM TARpolyA RNA) was added to a clean vial. Next, 2.5μL 1M(PH 7.5) HEPES was added into the vial. Then, 20.3μL DEPC-H$_2$O (or 28.7μL for TARpolyA RNA) was added. The temperature of the mixture was then raised to 80˚C for 2 minutes, followed by 60˚C for another 2 minutes using a water bath. Finally, 5μL 0.1M MgCl$_2$ was added into the vial. Next, the mixture was kept at 37˚C for 5 minutes followed by 0˚C with ice for 30 minutes. The RNA could then be used immediately or stored at 4˚C for later use up to a week. After applying the refolding protocol, 30μM RNAs were diluted to 0.5μM using the same HEPES buffer before AFM imaging. Brain PI(4,5)P2 purchased in powder form was dissolved in distilled water to 1 mM concentration

before use. For mixtures, the mixed solutions were obtained such that the final concentration of each component was 0.5μM.

## Mica substrates

Atomically smooth mica substrates were used in all AFM imaging experiments reported here. Mica substrates with 1cm diameter were obtained from Ted Pella Inc. (Redding, CA, USA). The scotch tape technique was used to obtain freshly cleaved surfaces used in all experiments. The surface roughness was measured to be 0.1~0.2nm. The clean mica surface is negatively charged with charge density $\sigma$ = -0.33C/m$^2$ in air and $\sigma$ = -2.5mC/m$^2$ in water [41–43]. Therefore, freshly cleaved raw mica is a perfect substrate for AFM imaging of HIV GagΔP6 because of its positive charge. However, as both ΨRNA and TARpolyA RNA are negatively charged they cannot be observed directly on the raw mica substrate. To overcome the repulsion between mica and RNAs, the mica surface was made positive by functionalization with APTES (3-aminopropyltriethoxy silane). The procedure of preparing APTES-treated mica was as follows. First, double-sided tape was used to stick a 10mm in diameter mica upon an AFM metal specimen disc with a diameter of 15mm. Second, Scotch tape was used to cleave mica until the mica surface was complete and flat. Then, 100μL APTES was added into a small plastic petri dish and put at the bottom of a desiccator with a plastic net onto which the freshly cleaved mica was placed. The dessicator was next evacuated with a mechanical vacuum pump. The vacuum suction was maintained for 30 minutes to allow APTES to evaporate. APTES treated mica was ready to use immediately or can be stored in a covered petri dish for later use. 30~50μL of the desired sample solution was next deposited on the freshly cleaved mica (or APTES treated mica for RNAs) before AFM imaging. When imaging the various complexes GagΔP6 formed, we always mixed the components in the appropriate molar ratio and then incubated the solution for 3 hours in order to confirm that the different distributions of the complexes have reached equilibrium before the solution was introduced to the AFM fluid cell containing the mica substrate. In addition, the same freshly cleaved mica substrate was used for the investigating all the Gag complexes: GagΔP6, GagΔP6/RNA, GagΔP6/ PI(4,5)P2 and GagΔP6/RNA/ PI(4,5)P2. Thus it is valid to make a comparison of the changes in the morphologies or population statistics of monomers, dimers and tetramers with the addition of RNA or PI(4,5)P2 to GagΔP6 in solution. During imaging on the mica substrate, no dynamical evolutions of the complexes were observed."

## AFM

Calibration of the AFM probes had to be done before imaging. The size of the AFM cantilever tip was similar to that of the proteins or protein complexes studied. The comparable tip size will lead to feature broadening, which is a common type of widely known convolution effect [44]. The major factors with respect to the feature broadening are the pyramidal geometry and curvature radius of the tip. The AFM probe used (HI'RES-C19/CR-AU, MikroMasch USA, Watsonville, CA, USA) was 125μm long and 22.5μm wide with a spring constant of 0.5N/m. it had a nominal resonant frequency of 65kHz in air and a ~32.36kHz in liquid. The special tips used had high aspect ratio and small tip radius. Nevertheless, the tip size still had to be taken into account when analyzing the sizes of HIV GagΔP6, RNAs, lipids and their mixed complexes. The calibration was done as follows. 2nm gold spheres were used to calibrate AFM probes due to the comparability of the heights of HIV GagΔP6 and two RNAs. Because the measured sample size also depends on the height of the sample, the actual tip size is given by

Eq (1) (see Supplementary Material for more details):

$$\lambda = L - 1.46D \tag{1}$$

Where $\lambda$ is the actual diameter of the AFM cantilever tip, $L$ is the measured size of the sample, and $D$ is the height of the sample. The effective tip diameter $t$ for any sample is the difference between measured size of the calibration standard sample and the height of the sample, as given by Eq (2) (see Supplementary Material for more details):

$$t = L - D = \lambda + 0.46D \tag{2}$$

For the 2nm gold spheres, after fitting to a normal distribution, the mean measured size $L = 7.56 \pm 0.09nm$, the mean height $D = 2.10 \pm 0.02nm$ which is the actual diameter of the 2nm gold particle, as shown in Fig 2. The total number of samples was 419 and the experiment was repeated twice. Therefore, the actual diameter is $\lambda = 4.5nm$ according to Eq (1). The effective tip size for HIV GagΔP6 and other GagΔP6 complexes, including GagΔP6-ΨRNA, GagΔP6-PI(4,5)P2, PI(4,5)P2-ΨRNA-GagΔP6, is $t_{Gag} = 5.4nm$ given that the measured height of HIV GagΔP6 is 1.9nm according to Eq (2). Similarly, the effective tip size for ΨRNA and TARpolyA RNA is $t_{RNA} = 5.0nm$ given that the measured height of both RNAs is 1.1nm.

The AFM Nanoscope IIIa (Veeco Metrology, Santa Barbara, CA, USA) was used in tapping mode. The procedure of the AFM operation in tapping mode in liquid environment is as follows [45]. First, a freshly cleaved mica (or APTES treated mica for RNAs) was mounted on the AFM metal specimen holder using double-sided tape. Next, a 30~50μL drop of following sample solutions used in the experiment was deposited on the mica: (I) ΨRNA (0.5μM), (II) TARpolyA RNA (0.5μM), (III) GagΔP6 (0.5μM), (IV) mixture of PI(4,5)P2-DPhPC-POPC (0.5μM : 5μM : 5μM) complex, (V) mixture of GagΔP6-ΨRNA (0.5μM : 0.5μM) complex, (VI) mixture of GagΔP6-TARpolyA RNA (0.5μM : 0.5μM) complex, (VII) mixture of GagΔP6-PI(4,5)P2 (0.5μM : 0.5μM) complex, (VIII) and mixture of PI(4,5)P2-ΨRNA-GagΔP6 (0.5μM : 0.5μM : 0.5μM) complex. Next the cantilever probe was mounted into the fluid cell. Care was taken to make sure there are no bubbles. The tip is completely immersed in the solution. This is critical for laser alignment when operating the AFM in a liquid environment. Next the laser signal was aligned and then the piezo was oscillated through a range of frequencies till the resonance frequency was found. Next the best resolution was obtained by adjusting the following scanning parameters: the vertical range, samples/line, scan size, scan rate, integral gain, proportional gain and amplitude setpoint etc. Additional checks were made after engaging the cantilever. During imaging the amplitude setpoint was adjusted such that the trace and retrace lines were matched. The force exerted on the sample should be as small as possible to prevent samples from being damaged. All the aforementioned parameters had to be adjusted collectively to achieve the best resolution.

## Results and discussion

As RNAs have been well studied with the AFM [46–50], they were used to benchmark the experiments reported here. The self-assembly of the CA domains in Gag and its mechanical properties have been recently investigated on a mica substrate with the AFM [51]. The CA domain was found to assemble in a hexagonal lattice and have self repair capacity after damage was induced [51]. In the AFM experiments here, we started with size and morphology measurements of the RNAs (ΨRNA and TARpolyA RNA) to benchmark and validate the measurement and analysis software developed. Independent calibration of the measurement resolution was done using 2nm Au spheres as discussed above. After confirmation of the validity of the technique we performed measurements on GagΔP6 and its complexes with RNAs

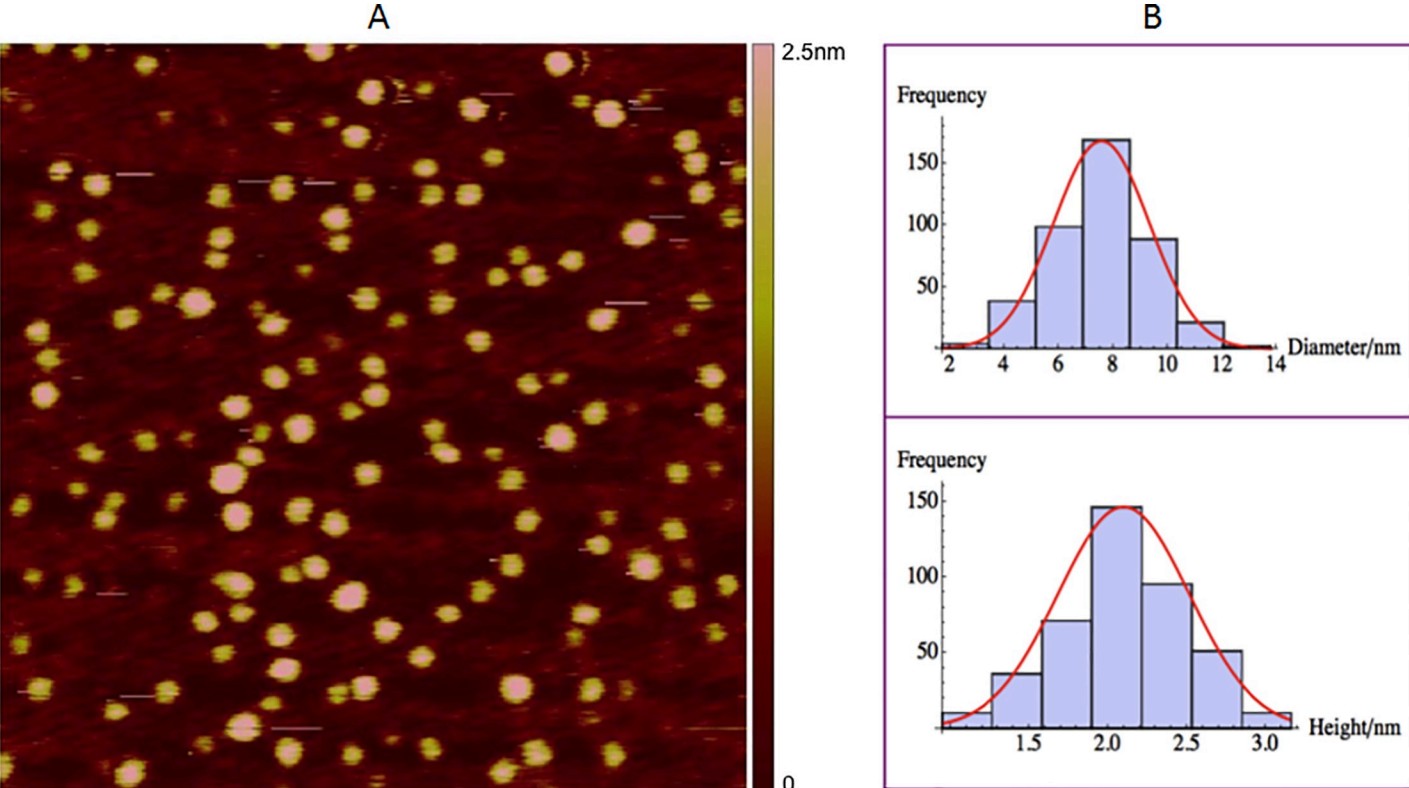

**Fig 2. Calibration of AFM cantilever tip.** AFM tip resolution calibration with 2nm diameter Au Sphere. (A) A typical AFM image of 2nm Au sphere on mica in tapping mode in liquid. The scan size is 250nm×250nm. The height color bar scale is 2.5nm. (B) Histogram of 2nm diameter Au sphere for the measured size on the plane of the substrate (top) and the measured height (bottom). Shown in red are normal distribution fits to the peaks. The mean measured size on the plane of the substrate is 7.56 ± 0.09 nm, and the mean height is 2.10 ± 0.02nm. The height is consistent with the sphere diameter. The size in the plane of the substrate reflects the role of the tip size. Please see text and supplemental materials section for more details. The total number of samples was 419 and the experiment was repeated twice.

and PI(4,5)P2 lipid. The summary of the results is provided in Table 1. All the experiments were repeated twice. Below we present the results from each individual experiment and also discuss the effect of the RNA and PI(4,5)P2 lipid interaction with GagΔP6.

## ΨRNA size and morphology measurements

ΨRNA (0.5µM) being negatively charged was imaged on positively charged APTES treated mica. Another motivation for measuring the ΨRNA by itself is to understand its individual morphology for future comparison with that observed in the various GagΔP6 complexes. A typical AFM image of ΨRNA was shown in Fig 3A. In Fig 3A, the AFM image of ΨRNA shows that most of ΨRNA molecules seem to have inverted "L" shape. As shown in Fig 3B and in Table 1, the mean height is 1.10 ± 0.01nm. This height is in between the values 0.5nm, 2.5, 2.6nm found from double-stranded RNA [28,48,52] and similar to 0.9~1.2nm that found by Hansma et al. [53]. The lateral size of the image was analyzed using specially developed software analysis (see Supplementary Material for more details). The statistics of the length (longest dimension) and width (longest perpendicular dimension to the length) and height were plotted in Fig 3Bb. As can be observed there are two distinct peaks for the length and similarly two distinct peaks for the width. The three dimensional smooth histogram of the same length and width population distribution is shown in Fig 3C. Two distinct populations can be observed. The size of the first distribution with a mean length of 17.9 ± 0.2nm and width

**Table 1. Statistics of AFM measurement of GagΔP6, RNAs and their complexes.**

| | | Percentage | Length/nm | Width/nm | Height/nm |
|---|---|---|---|---|---|
| ψRNA | Monomer | 74% ± 2% | 17.9 ± 0.2 | 1.01 ± 0.02 | 1.10 ± 0.01 |
| | Dimer | 26% ± 2% | 34.6 ± 0.3 | 3.8 ± 0.1 | |
| TARpolyA RNA | Monomer | 100% | 17.1 ± 0.2 | 1.10 ± 0.05 | 1.10 ± 0.01 |
| GagΔP6 | Monomer | 59% ± 3% | 10.3 ± 0.1 | 6.2 ± 0.1 | 1.93 ± 0.01 |
| | Dimer | 35% ± 2% | 20.0 ± 0.2 | | |
| | Tetramer | 6% ± 1% | 29.0 ± 0.3 | 12.9 ± 0.2 | |
| GagΔP6-ψRNA | Monomer | 35% ± 2% | 10.6 ± 0.2 | 6.8 ± 0.1 | 1.90 ± 0.01 |
| | Dimer | 49% ± 2% | 22.4 ± 0.1 | | |
| | Tetramer | 16% ± 1% | 31.2 ± 0.2 | 13.8 ± 0.1 | |
| GagΔP6-TARpolyA RNA | Monomer | 58% ± 3% | 10.8 ± 0.1 | 6.2 ± 0.1 | 1.93 ± 0.01 |
| | Dimer | 37% ± 2% | 20.7 ± 0.2 | | |
| | Tetramer | 5% ± 1% | 29.9 ± 0.3 | 13.6 ± 0.2 | |
| GagΔP6-PI(4,5)P2 | Monomer | 47% ± 2% | 11.2 ± 0.1 | 6.7 ± 0.1 | 1.93 ± 0.01 |
| | Dimer | 41% ± 2% | 21.1 ± 0.1 | | |
| | Tetramer | 12% ± 1% | 30.4 ± 0.2 | 14.0 ± 0.2 | |
| PI(4,5)P2-ψRNA-GagΔP6 | Monomer | 17% ± 1% | 10.9 ± 0.3 | 7.4 ± 0.2 | 1.91 ± 0.01 |
| | Dimer | 51% ± 3% | 23.8 ± 0.2 | | |
| | Tetramer | 32% ± 2% | | 22.1 ± 0.2 | |

1.01 ± 0.02nm and height of 1.10 ± 0.01nm corresponds to the ΨRNA monomer. The second peak with a mean length of 34.6 ± 0.3nm and width 3.8 ± 0.1nm and height of 1.10 ± 0.01nm corresponds to the ΨRNA dimer. Typical images of the monomer and dimers are enclosed in red and green boxes respectively.

The expected size of the 109-nucleotide ΨRNA monomer can be calculated from the literature. RNAs most commonly adopt either A-form or A'-form conformation [54, 55]. A-form RNA has 11 nucleotide per helical pitch and A'-form has 12 nucleotide per helical pitch [55]. The rise per base pair for the A-form and A'-form are 0.38nm and 0.27nm, respectively [56]. Given that most of ΨRNA nucleotides are self-paired with each other as shown in Fig 1B, ΨRNA should be double-stranded with 55 base pairs. This leads to a length of 20.9nm and 14.9nm for the A-form and A'-form respectively. These values are consistent with the mean measured monomer length 17.9nm. The width of ΨRNA monomer is 1.01 ± 0.02nm consistent with the height of ΨRNA. For the monomer the width and heights will be the same as it is cylindrical in shape. The mean measured length of ΨRNA dimer is 34.6 ± 0.3nm, which is approximately twice as long as that of ΨRNA monomer. This means ΨRNA dimer consists of two ΨRNA monomers connecting head to head overlap. The width of ΨRNA dimer is roughly 3.8 ± 0.1nm that is much larger than two times the width of ΨRNA monomer. This is probably due to some overlap of the two RNA's on dimerization. This conclusion is reasonable based on the results from Refs. [30–32] and the potential overlap region is the palindromic DIS base pair region located in the SL1 loop of ΨRNA as shown in Fig 1B. Next the population distribution between monomers and dimers was analyzed. The length and width histograms in Fig 3B were fit to a normal distribution. Both histograms lead to a population distribution of 74% monomer and 26% dimer respectively.

## TARpolyA RNA size and morphology measurements

TARpolyA RNA (0.5μM) being negatively charged was also imaged on positively charged APTES treated mica. Similar to ΨRNA, the motivation for measuring TARpolyA RNA by itself

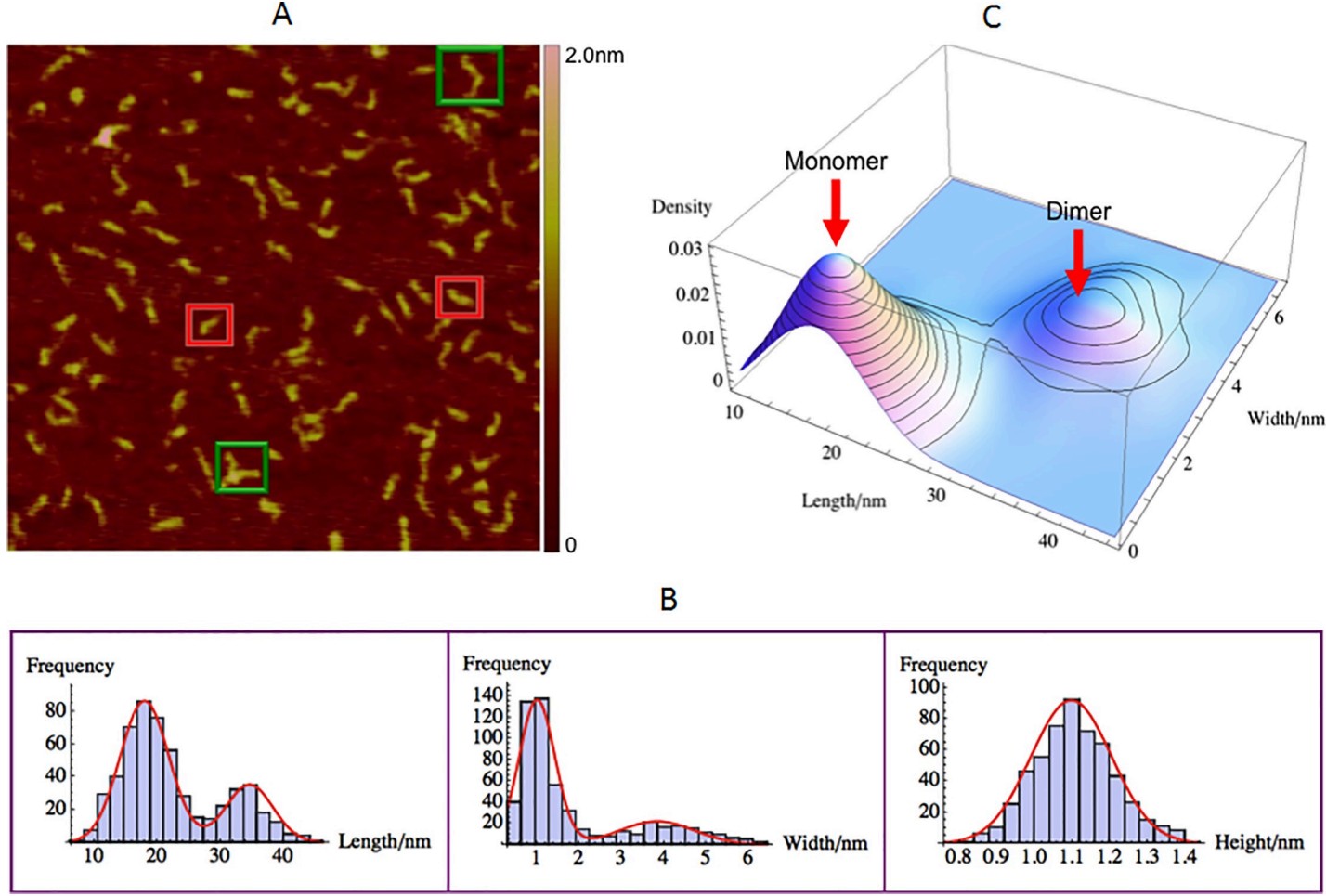

**Fig 3. ΨRNA size and morphology with AFM.** 0.5μM ΨRNA on positively charged mica(+). (A) A typical AFM image with a scan size of 500nm×500nm. The height color bar scale is 2.0 nm. A few characteristic ΨRNAs are shown in boxes: monomer (red) and dimer (green). (B) Histogram for length (left), width (middle), and height (right). Shown in red are normal distribution fits to the peaks. (C) Three dimensional smooth histogram, where red arrows indicate monomer and dimer. The mean height is 1.10 ± 0.01nm. The first peak with a mean length of 17.9 ± 0.2nm and width 1.01 ± 0.02nm corresponds to the ΨRNA monomer. The second peak with a mean length of 34.6 ± 0.3nm and width 3.8 ± 0.1nm corresponds to the ΨRNA dimer. The total number of samples was 551 and the experiment was repeated twice.

is to benchmark the experiments and analysis protocol as well as understand its individual morphology for comparison with that observed in the various GagΔP6 complexes. The observed typical AFM image of TARpolyA RNA in the buffer solution is shown in Fig 4. In Fig 4A, TARpolyA RNA image shows that most of TARpolyA RNA molecules seem to have inverted "L" shape just like ΨRNA. Some examples are shown enclosed in red boxes. The size distribution of the length (longest dimension), width (longest perpendicular dimension to the length) and the height are shown in Fig 4B. As shown in Fig 4B and in Table 1, the mean height of TARpolyA RNA is 1.10 ± 0.01nm. This height is consistent with that of ΨRNA and expectations from structure. In Fig 4B and 4C we observe only one peak in the length and width distributions. The mean length of the TARpolyA RNA monomer is 17.1 ± 0.2nm. This value is slightly less than that of the ΨRNA monomer observed earlier. Similar to ΨRNA, TARpolyA RNA should also be doubled-stranded as shown in Fig 1(A). Thus the slightly smaller length of 0.8nm is reasonable from the 5 fewer nucleotides, given that ΨRNA contains 109 nucleotides while TARpolyA RNA has only 104 nucleotides. It is noteworthy that the width histogram distribution of TARpolyA RNA is 1.10 ± 0.05nm which is the same as the height and consistent

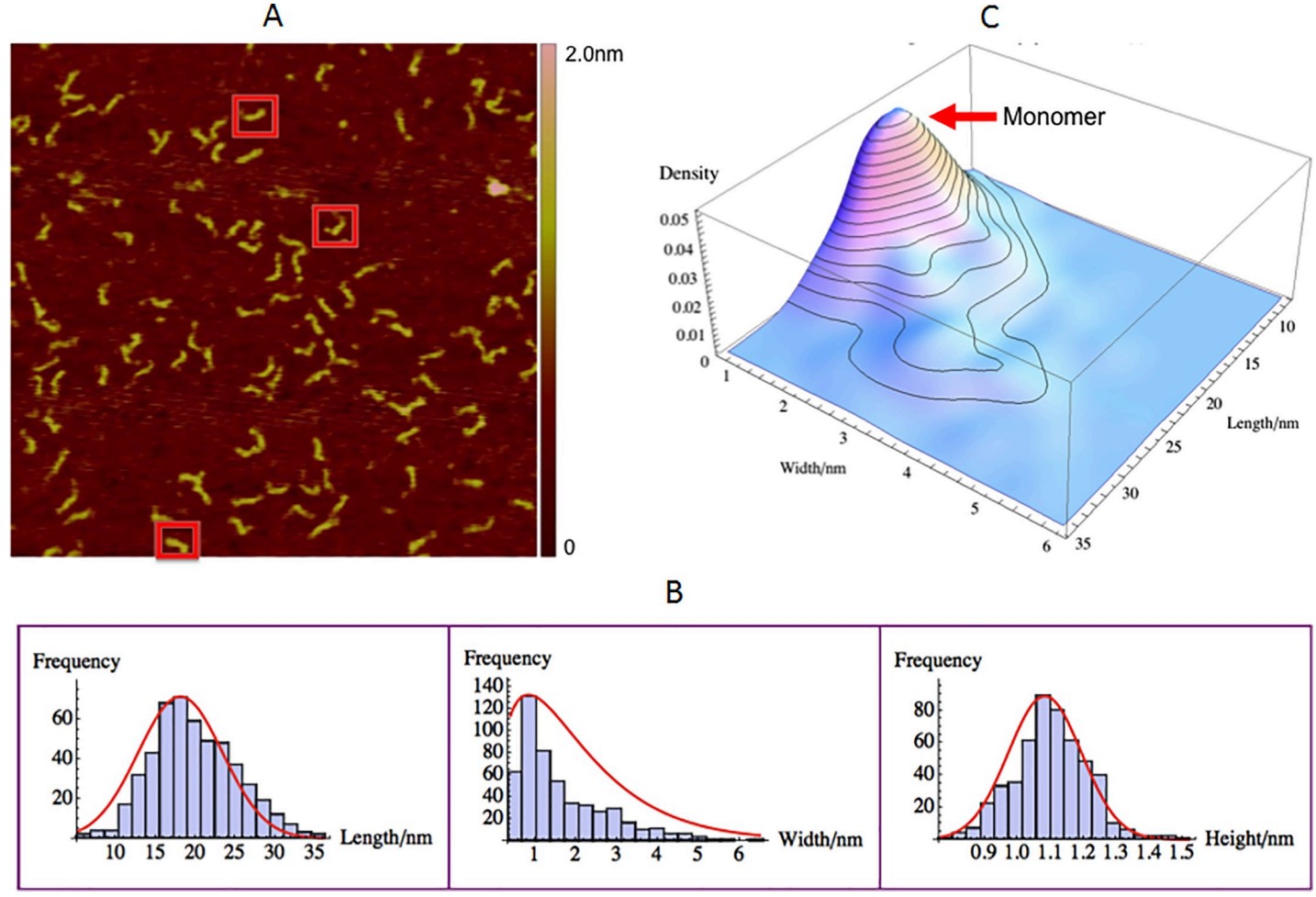

**Fig 4. TARpolyA RNA size and morphology with AFM.** 0.5μM TARpolyA RNA on positively charged mica(+). (A) AFM image with a scan size of 500nm×500nm. The height color bar scale is 2.0nm. A few characteristic TARpolyA RNAs are boxed in red. (B) Histogram for length (left), width (middle), and height (right). Shown in red are normal distribution fits to the peaks for length and height. Width is fit to a gamma distribution due to its non-negativity and skewness. (C) Three dimensional smooth histogram, where the red arrow indicates the monomer distribution. The peak corresponding to the TARpolyA RNA monomer has a mean length of 17.1 ± 0.2nm, width of 1.10 ± 0.05nm and height of 1.10 ± 0.01nm. The total number of samples was 504 and the experiment was repeated twice.

with the cylindrical structure for TARpolyA RNA. As the length and width distribution have only one observable peak, it is concluded that the TARpolyA exists in solution predominantly as a monomer. This is unlike that observed with ΨRNA where 26% dimers were found.

### GagΔP6 size and morphology measurements

The morphology of GagΔP6 (0.5μM), being net positive charge, was measured using the AFM on a freshly negatively charged mica(-) substrate in solution. A typical AFM image is shown in Fig 5A. The motivation for measuring GagΔP6 is to serve as a control before addition of RNAs and lipids. Some characteristics of common shapes observed were shown in red, green and blue boxes. In Fig 5A, the AFM image of GagΔP6 shows that most of GagΔP6 molecules have ellipsoidal shape rather than the expected rod-like shape of the extended molecule.

As shown in Fig 5B and Table 1, the mean height is 1.93 ± 0.01nm which is consistent with the expectation that the diameter of Gag is around 2~3nm when it is in the extended form as reported in Ref. [4]. The length and width of the observed images are plotted in Fig 5B. As

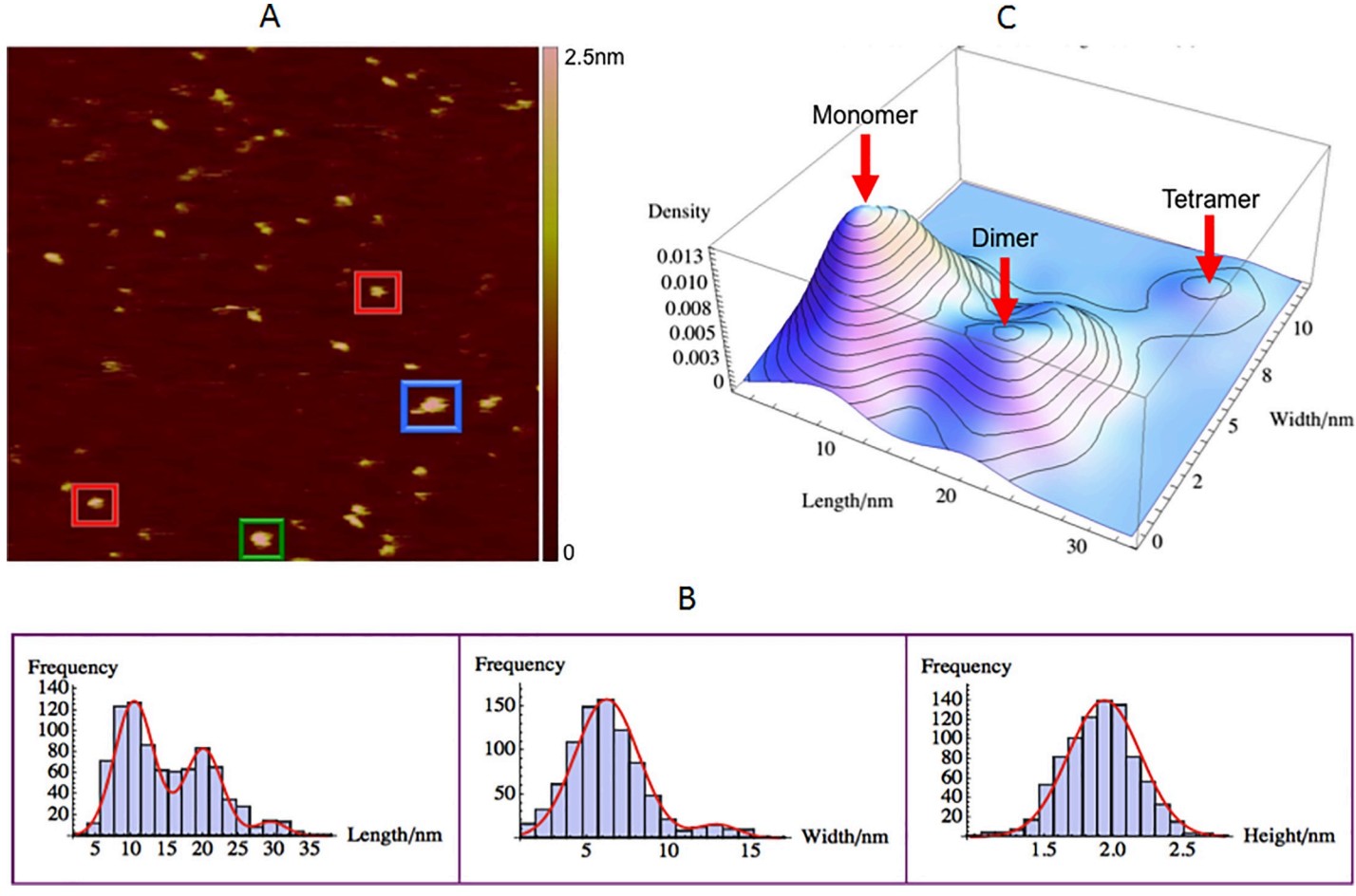

**Fig 5. GagΔP6 size and morphology with AFM.** 0.5μM GagΔP6 on negatively charged mica(-). (A) A typical AFM image with a scan size of 500nm×500nm. The height color bar scale is 2.5nm. A few characteristic GagΔP6s are boxed: monomer (red), dimer (green) and tetramer (blue). (B) Histograms for length (left), width (middle), and height (right). Shown in red are normal distribution fits to the peaks. (C) Three dimensional smooth histogram, where red arrows indicate the monomer, dimer, and tetramer distributions. The mean height is 1.93 ± 0.01nm for all three. The first peak with a mean length of 10.3 ± 0.1nm and width 6.2 ± 0.1nm corresponds to the GagΔP6 monomer. The second peak with a mean length of 20.0 ± 0.2nm and width 6.2 ± 0.1nm corresponds to the GagΔP6 dimer. The third peak with a mean length of 29.0 ± 0.3nm and width 12.9 ± 0.2nm corresponds to the GagΔP6 tetramer. The total number of samples was 858 and the experiment was repeated twice.

observed three peaks were observed in the length distribution but only two peaks in the width distribution. Normal distributions were fit to the length histogram to find peak mean values of 10.3 ± 0.1nm, 20.0 ± 0.2nm and 29.0 ± 0.3nm. Similarly the mean values of the two width distributions were found to be 6.2 ± 0.1nm and 12.9 ± 0.2nm respectively.

It might seem counterintuitive to have three distinct peaks for length while having only two distinct peaks for the width. To have more insight, a three dimensional smooth histogram of the same length-width data was plotted as shown in Fig 5C. The shortest length and width distribution would correspond to that of the monomer and the distribution with the next larger length would be expected to correspond to that of the dimer. From Fig 5C the monomer and dimer have the same width. Thus in the width distribution histogram they are represented together and correspond to the first peak at 6.2 ± 0.1nm. The statistical analysis of the population also confirmed this assumption as the first peak of the width histogram is the sum of monomer and dimer populations in the two peaks of the length histogram as given in Table 1. This analysis was done by fitting each length and width peak to a normal distribution. From the length distribution, the percentages of monomer, dimer, and tetramer are 59%, 35%, and

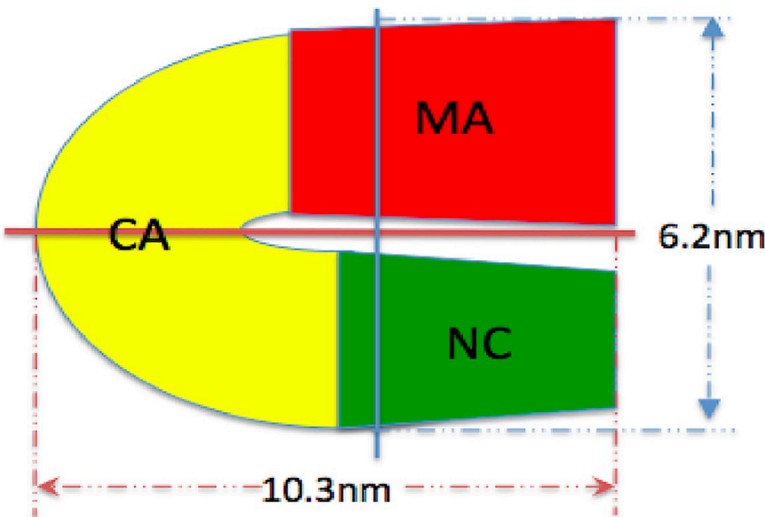

**Fig 6. Model of GagΔP6 morphology.** Rough model of the GagΔP6 monomer based on the measured mean values of the length and width. MA domain is in red, CA domain is in yellow, and NC domain is in green.

6%, respectively. The population observed in the monomer and dimer from the length adds up to that observed in the first peak of the width distribution.

Previous studies using hydrodynamic and neutron scattering measurements showed HIV GagΔP6 is supposed to adopt a compact conformation such that MA and NC domains of GagΔP6 are close to each other even though both of them are positively charged [3–5, 57]. The hydrodynamic radius, $R_h$, of GagΔP6 given by three different hydrodynamic tests are 3.6nm, 3.8nm, and 4.1nm, respectively. The radius of gyraton, $R_g$, of GagΔP6 is best estimated to be 3.4nm from small angle neutron scattering (SANS). The $R_g$ of GagΔP6 when it is a 25nm straight rod is supposed to be 7.2nm [57]. The average $R_g$ of GagΔP6 in solution measured by SANS is also a monotonically increasing function of the GagΔP6 concentration, with maximum of $R_g$ = 5nm at extremely high concentration, which means GagΔP6 molecules are in monomer-dimer equilibrium [57].

Based on the AFM studies presented above, we can project approximate confirmations based on the various lengths and widths of the populations observed. For the case of the monomer given a length of 10.3nm and width of 6.2nm a potential "C" like shape can be conjectured. A schematic of a potential monomer structure is shown in Fig 6. In neutron scattering and hydrodynamic experiments [4] the Gag was observed to be folded over, with its N-terminal MA domain near its C- terminal NC domain in three-dimensional space. The conjectured model in Fig 6 is consistent with the measurements of the folded Gag monomer expected in solution and the AFM size measurements. The conjectured model in Fig 6 is also consistent with that of a fully extended Gag having a rod-like shape of 2 nm radius and 20–30 nm length postulated in the literature [2]. For the case of the Gag dimer, based on the AFM measured dimensions of a length of 20.0 nm and width of 6.2 nm (same as monomer) a model of the GagΔP6 dimer would have two "C" shaped monomers connected back to back through their CA-CA domain interaction [4]. The CA-CA domain interaction leading to dimerization has been reported in the literature [4]. The last population size distribution of GagΔP6 in Fig 5(C) has a length of 29.0nm and width is 12.9nm. This population from gel electrophoresis corresponds to that of a tetramer (see Supplemental Material). From the measured size of the tetramers here, they could be potentially formed by the interaction of two dimers.

## GagΔP6-ΨRNA complex size, morphology and interaction

AFM measurements of the GagΔP6-ΨRNA (0.5μM : 0.5μM) complex were done on negatively charged mica(-). The motivation for measuring GagΔP6-ΨRNA complex is to investigate the effect of addition of specific ΨRNA to GagΔP6. According to current models the NC domain binds with ΨRNA and this interaction is specific and is a critical step in the formation of HIV [22]. Fig 7A is a typical AFM image of GagΔP6-ΨRNA complex. As shown in Fig 7B, the mean height of GagΔP6-ΨRNA complex is 1.90 ± 0.01nm which is roughly the same as the height of just GagΔP6 discussed earlier. This height is consistent with expectation given that the height of GagΔP6 and ΨRNA are 1.93nm and 1.10nm, respectively. The length and width histograms of the complexes observed are shown in Fig 7B. Similar to GagΔP6, GagΔP6-ΨRNA also has three peaks for length and two peaks for width. Normal distributions were fit to the length histogram to find peak mean values of 10.6 ± 0.2nm, 22.4 ± 0.1nm and 31.2 ± 0.2nm. Similarly the mean values of the two width distributions were found to be 6.8 ± 0.1nm and 13.8 ± 0.1nm respectively. The size of the length and widths are slightly larger than that

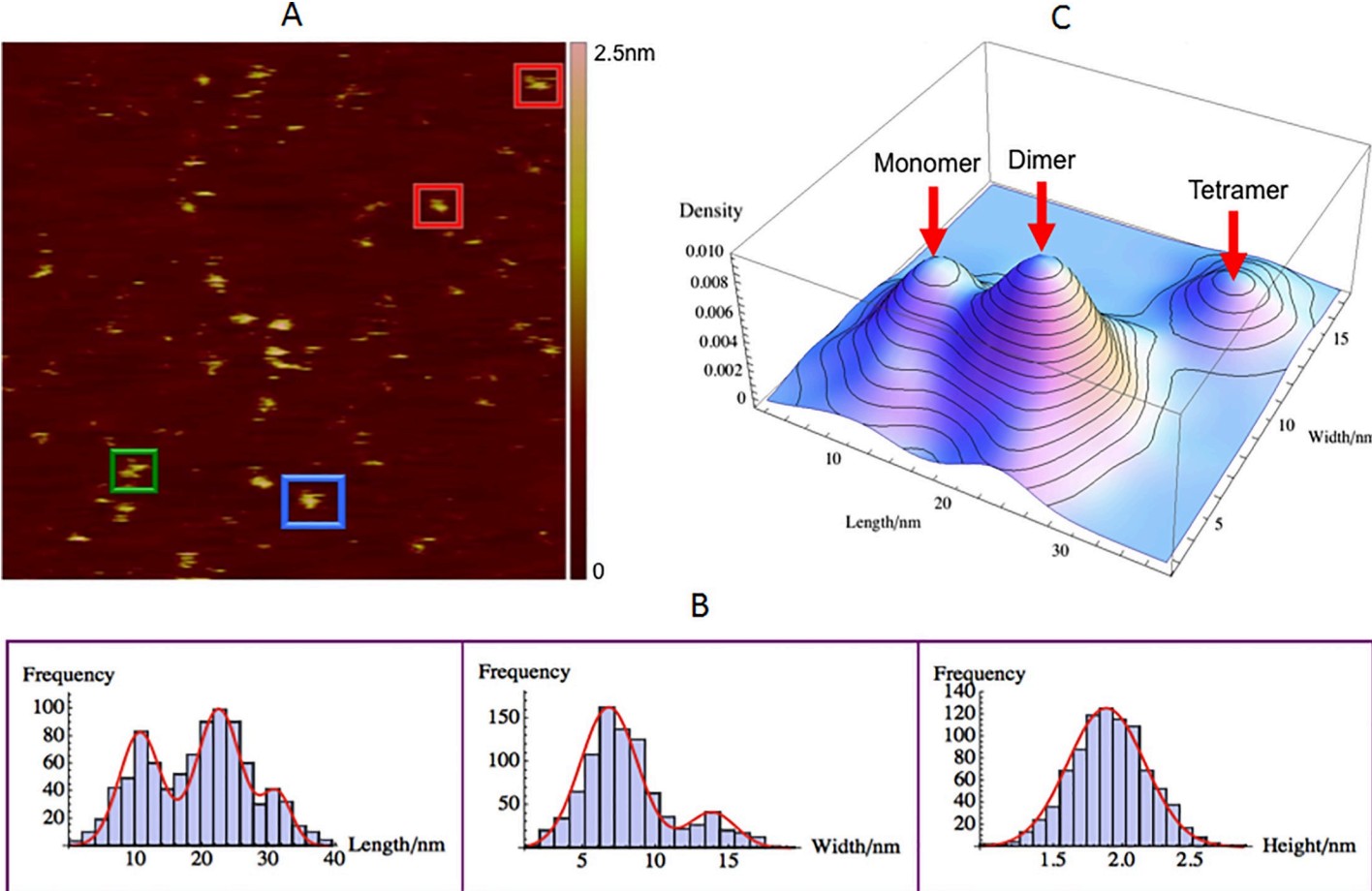

**Fig 7. GagΔP6-ΨRNA interaction complex size and morphology.** The mixture of GagΔP6-ΨRNA (0.5μM : 0.5μM) complex on negatively charged mica(-). (A) A typical AFM image with a scan size of 500 nm×500 nm. The height color bar scale is 2.5nm. A few characteristic GagΔP6-ΨRNA complexes are boxed: monomer (red), dimer (green) and tetramer (blue). (B) Histogram for length (left), width (middle), and height (right). Shown in red are normal distribution fits to the peaks. (C) Three dimensional smooth histogram, where red arrows indicate monomer, dimer, and tetramer. The mean height is 1.90 ± 0.01nm for all three complexes. The first peak with a mean length of 10.6 ± 0.2nm and width 6.8 ± 0.1nm corresponds to the GagΔP6-ΨRNA monomer. The second peak with a mean length of 22.4 ± 0.1nm and width 6.8 ± 0.1nm corresponds to the GagΔP6-ΨRNA dimer. The third peak with a mean length of 31.2 ± 0.2nm and width 13.8 ± 0.1nm corresponds to the GagΔP6-ΨRNA tetramer. The total number of samples was 895 and the experiment was repeated twice.

found with only GagΔP6. This is consistent with attachment of ΨRNA of around 1nm to the GagΔP6.

To understand the role of the ΨRNA addition the three-dimensional population distribution of the length and width as shown in Fig 7C was analyzed. As with GagΔP6, three peaks corresponding to monomer, dimer and tetramer complexes were observed. However the population distributions of the monomer, dimer and tetramer are different for the GagΔP6-ΨRNA complex. Here we observed 35% monomer, 49% dimers and 16% tetramers. In contrast with only GagΔP6 we observed 59% monomers 35% dimers and 6% tetramers. The monomer decreased by 24%, the dimer increased by 14% and the tetramer increased by 10%. Thus the addition of ΨRNA promotes multimerization of the GagΔP6 leading to higher populations of dimers and tetramers.

The length of monomer remained roughly the same as prior to the addition of ΨRNA. The lengths of dimer and tetramer were increased by about 2nm. The width increased by about 0.5nm~1nm for the monomer, dimer, and tetramer. This is consistent with the addition of ΨRNA of around 1nm to this dimension. The most important conclusion of the effect of the addition of ΨRNA to GagΔP6 is that ΨRNA can bind with GagΔP6 and facilitate GagΔP6 multimerization given the increases in percentages of dimer and tetramer population.

## GagΔP6-TARpolyA RNA complex size, morphology and interaction

GagΔP6-TARpolyA RNA (0.5μM : 0.5μM) complex was measured on a negatively charged mica(-) surface. The motivation for measuring GagΔP6-TARpolyA RNA complex is to verify the effect of the addition of non-specific TARpolyA RNA to GagΔP6 and compare it to the addition of specific ΨRNA discussed above. Fig 8A is a typical AFM image of GagΔP6-TARpolyA RNA complex. As shown in Fig 8B, the mean height of GagΔP6-ΨRNA complex is 1.93 ± 0.01nm.

To understand the role of the TARPolyA RNA interaction with GagΔP6 the three dimensional population distribution of the length and width as shown in Fig 8C was analyzed. As with GagΔP6, three peaks corresponding to monomer, dimer and tetramer complexes were observed. However in contrast to the case of the GagΔP6-ΨRNA complex in Fig 7C the population distribution of monomer, dimer and tetramer are very similar to that of GagΔP6 alone observed in Fig 5C. The sizes of GagΔP6-TARpolyA RNA complex monomer, dimer and tetramer were found slightly larger than that of GagΔP6 alone. Thus the addition of TARPolyA RNA might interact with GagΔP6 but does not promotes multimerization of the GagΔP6 as observed with ΨRNA. These experiments were repeated and the results were always reproducible. Webb et al. reported HIV Gag can bind with both ΨRNA and TARpolyA RNA but with distinct binding mechanisms [22]. They proposed that HIV GagΔP6 binds with TARpolyA RNA through both MA and NC domains whereas ΨRNA binds only through NC domain and leaves MA domain free to later interact with the lipid membrane. Other studies also showed that HIV GagΔP6 can bind with both ΨRNA and non-Ψ RNAs but the selective binding with ΨRNA is more energetically favorable than other non-Ψ RNAs for HIV virus assembly [58–60]. The role of the RNA in multimerization of the Gag was not addressed in these studies. The conclusion based on our data is that it is highly likely that HIV GagΔP6 interacts with ΨRNA and TARpolyA RNA through different mechanisms such that ΨRNA facilitates GagΔP6 multimerization while TARpolyA RNA does not.

## GagΔP6-PI(4,5)P2 Complex Size, Morphology and Interaction

The GagΔP6-PI(4,5)P2 (0.5μM : 0.5μM) complex was measured on negatively charged mica (-). The Gag and lipid were diluted to 1μM before mixing. Then, equal amounts of Gag and

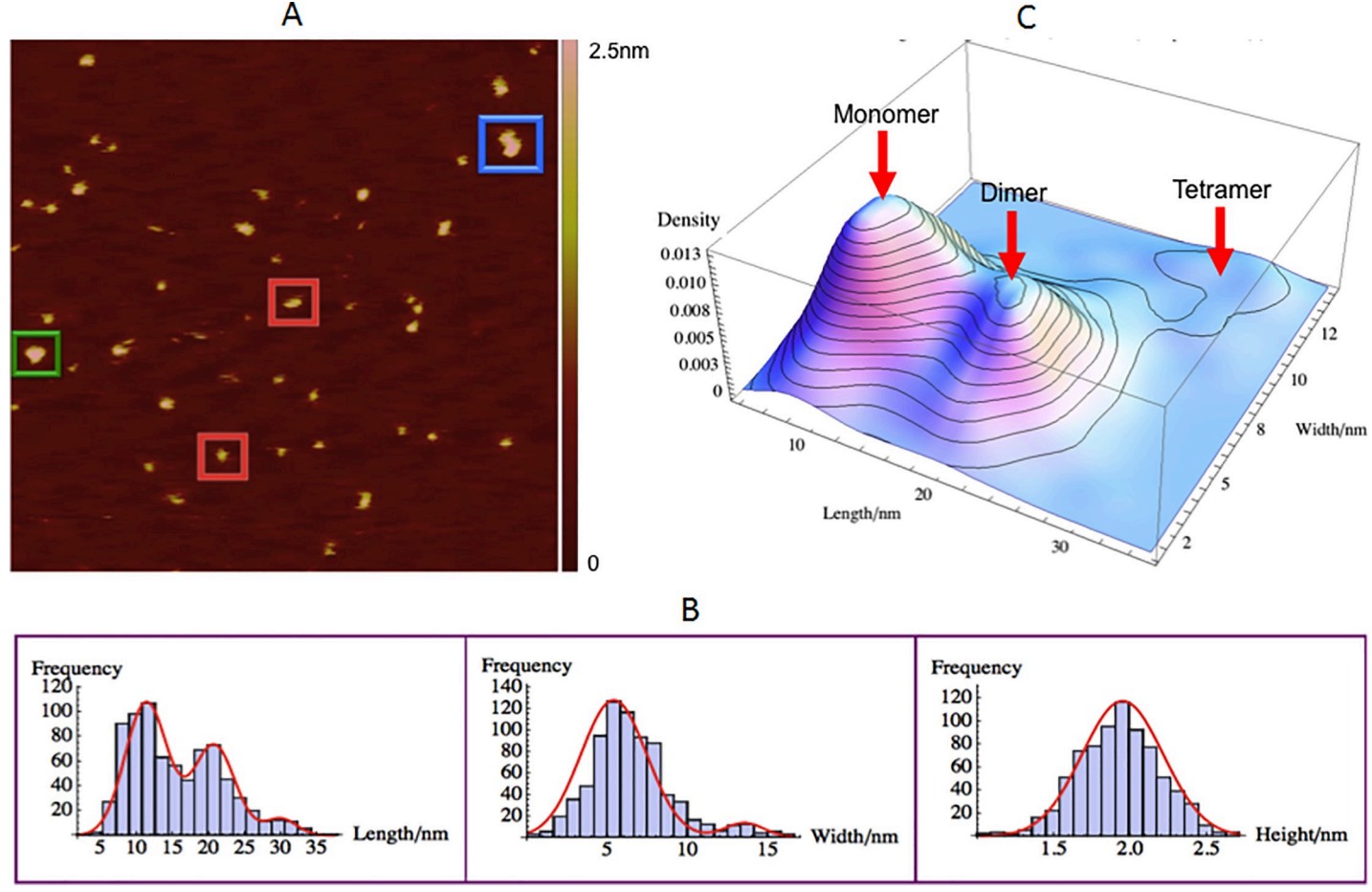

**Fig 8. GagΔP6-TARpolyA RNA interaction complex size and morphology.** Mixture of GagΔP6-TARpolyA RNA (0.5μM : 0.5μM) complex on negatively charged mica(-). (A) A typical AFM image with a scan size of 500nm×500nm. The height color bar scale is 2.5nm. A few characteristics TARpolyA RNA complexes are boxed: monomer (red), dimer (green) and tetramer (blue). (B) Histogram for length (left), width (middle), and height (right). Shown in red are normal distribution fits to the peaks. (C) Three dimensional smooth histogram, where monomer, dimer, and tetramer are indicated by red arrows. The mean height is 1.93 ± 0.01nm for all three complexes. The first peak with a mean length of 10.8 ± 0.1nm and width 6.2 ± 0.1nm corresponds to the GagΔP6-TARpolyA RNA monomer. The second peak with a mean length of 20.7 ± 0.2nm and width 6.2 ± 0.1nm corresponds to the GagΔP6-TARpolyA RNA dimer. The third peak with a mean length of 29.9 ± 0.3nm and width 13.6 ± 0.2nm corresponds to the GagΔP6-TARpolyA RNA tetramer. The total number of samples was 766 and the experiment was repeated twice.

lipid were mixed. The AFM measurement was performed after the mixture was incubated for 3 hours. A typical AFM image is shown in Fig 9A. The motivation for measuring GagΔP6-PI (4,5)P2 complex is to explore the effect of addition of lipid PI(4,5)P2 to GagΔP6. The current understanding is that both MA and NC domains of Gag can bind to PI(4,5)P2 through electrostatic forces.

As shown in Fig 9B(b), the mean height of GagΔP6-PI(4,5)P2 complex is 1.93 ± 0.01nm that is roughly the same as the height of just GagΔP6. This height is consistent with expectation given the small size of PI(4,5)P2 in comparison to GagΔP6. The length and width histograms of the complexes observed are shown in Fig 9B. Similar to GagΔP6, GagΔP6-PI(4,5)P2 also has three peaks for length and two peaks for width. Normal distributions were fit to the length histogram to find peak mean values of 11.2 ± 0.1nm, 21.1 ± 0.1nm and 30.4 ± 0.2nm. Similarly the mean values of the two width distributions were found to be 6.7 ± 0.1nm and 14.0 ± 0.2nm respectively. The size of the length and widths are slightly larger than that with only GagΔP6.

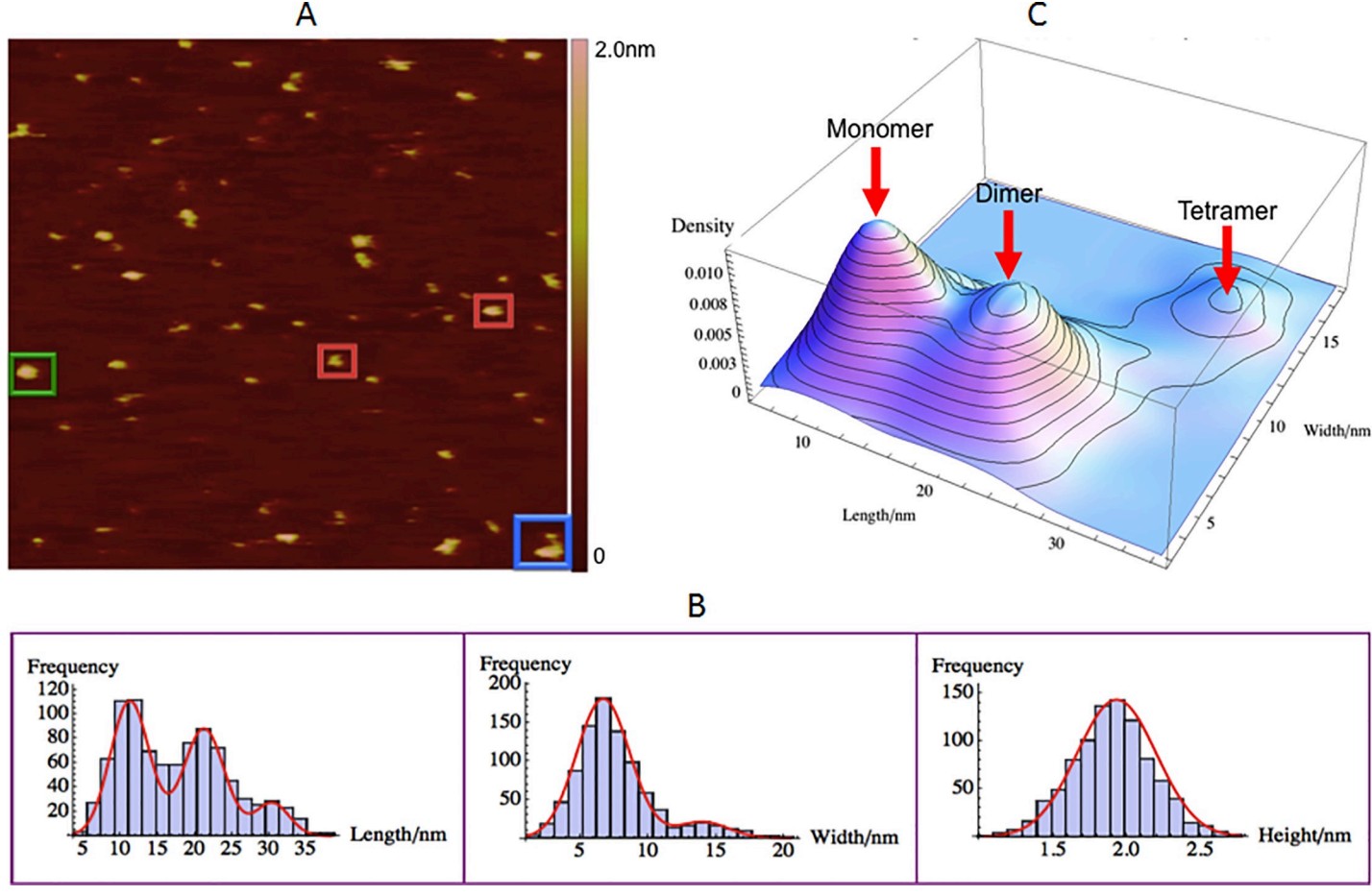

**Fig 9. GagΔP6-PI(4,5)P2 interaction complex size and morphology.** Mixture of GagΔP6-PI(4,5)P2 (0.5μM : 0.5μM) complex on negatively charged mica(-). (A) A typical AFM image with a scan size of 500nm×500nm. The height the color bar scale is 2.5nm. A few characteristic GagΔP6-PI(4,5)P2 complexes are boxed: monomer (red), dimer (green) and tetramer (blue). (B) Histogram for length (left), width (middle), and height (right). Shown in red are normal distribution fits to the peaks. (C) Three dimensional smooth histogram, where monomer, dimer, and tetramer are indicated by red arrows. The mean height is 1.93 ± 0.01nm for all complexes. The first peak with a mean length of 11.2 ± 0.1nm and width 6.7 ± 0.1nm corresponds to the GagΔP6-PI(4,5)P2 monomer. The second peak with a mean length of 21.1 ± 0.1nm and width 6.7 ± 0.1nm corresponds to the GagΔP6-PI(4,5)P2 dimer. The third peak with a mean length of 30.4 ± 0.2nm and width 14.0 ± 0.2nm corresponds to the GagΔP6-PI(4,5)P2 tetramer. The total number of samples was 903 and the experiment was repeated twice.

In comparison to the GagΔP6-ΨRNA complex the GagΔP6-PI(4,5)P2 complex has monomers of slighter larger length while the dimers and tetramers are of slightly smaller length.

To understand the role of PI(4,5)P2 addition to GagΔP6 the three dimensional population distribution of the length and width as shown in Fig 9C was analyzed. As with GagΔP6, three peaks corresponding to monomer, dimer and tetramer complexes were observed. However the population distribution of monomer, dimer and tetramer are different from that observed with GagΔP6 alone or for the GagΔP6- ΨRNA complex. Here we observed 47% monomer, 41% dimers and 12% tetramers. In contrast with only GagΔP6 we observed 59% monomers 35% dimers and 6% tetramers and for the GagΔP6- ΨRNA complex we observed 35% monomer, 49% dimers and 16% tetramers. Thus the addition of PI(4,5)P2 promotes multimerization of the GagΔP6. The increases in percentages of dimer and tetramer indicate that PI(4,5)P2 can bind with GagΔP6 as reported in other studies using confocal microscopy, nuclear magnetic resonance and equilibrium flotation assay [35, 40, 61–65]. But the increase in GagΔP6 multimerization with the addition of PI(4,5)P2 is less than that observed with ΨRNA. The

length and width of monomer, dimer, tetramer all increased by about 1nm. This is probably because of the size of PI(4,5)P2 attached to the ends of both MA and CA domains of HIV GagΔP6.

### PI(4,5)P2-ΨRNA-GagΔP6 complex size, morphology and interaction

The PI(4,5)P2-ΨRNA-GagΔP6 (0.5μM : 0.5μM : 0.5μM) complex was measured on a negatively charged mica(-). According to the prevailing model the lipid interacts with the MA domain of GagΔP6-ΨRNA complex leading to a conformational change [22]. In these experiments to study the PI(4,5)P2-ΨRNA-GagΔP6 complex, PI(4,5)P2 and ΨRNA were first mixed together in solution followed by the addition of GagΔP6. This mixture was then used to measure the size and size distribution using the AFM.

Fig 10A is a typical AFM image of PI(4,5)P2-ΨRNA-GagΔP6 complex. As shown in Fig 10B, the mean height of PI(4,5)P2-ΨRNA-GagΔP6 complex is 1.91 ± 0.01nm that is roughly the same as the height of just GagΔP6. This height is consistent with expectation given that the height of GagΔP6 and ΨRNA-GagΔP6 discussed earlier. The height of PI(4,5)P2 is much smaller by comparison. The length and width histograms of the complexes observed are shown in Fig 10B. In contrast to GagΔP6, GagΔP6-ΨRNA and the GagΔP6- PI(4,5)P2 complexes there are only two peaks for length and two peaks for width. Normal distributions were fit to the length histogram to find peak mean values of 10.9 ± 0.3nm, and 23.8 ± 0.2nm. Similarly the mean values of the two width distributions were found to be 7.4 ± 0.2nm and 22.1 ± 0.2nm respectively. In comparison to the binary complexes studied earlier the monomer length is approximately similar, while the monomer width is slightly larger. For the dimer, the length is almost 60% larger than that of GagΔP6-ΨRNA and the GagΔP6- PI(4,5)P2 binary complexes.

To understand the PI(4,5)P2-ΨRNA-GagΔP6 complex the three dimensional population distribution of the length and width as shown in Fig 10C was analyzed. In contrast to all others above, here the dimer and tetramer have the same length. The width of the tetramer is much larger than all previous cases, increasing from 13-14nm to 22nm. Thus in the PI(4,5)P2-ΨRNA-GagΔP6 complex the GagΔP6 undergoes a dramatic conformational change. This is consistent with the GagΔP6 MA and NC domains moving further away from each other and taking on a rod-like confirmation. From the population distribution in Fig 10C, we observed 17% monomer, 51% dimers and 32% tetramers. In contrast, with only GagΔP6 were we observed 59% monomers 35% dimers and 6% tetramers, the multimerization has increased considerably. Comparing to the populations in the binary mixtures of ΨRNA-GagΔP6 and PI(4,5)P2-GagΔP6 were we observed 35% and 47% monomer, 49% and 41% dimers and 16% and 12% tetramers respectively, here in particular we observe much higher percentage of tetramers. Thus the addition of PI(4,5)P2 along with ΨRNA promotes not only the dimerization of the GagΔP6 but in particular the multimerization to tetramers and higher order complexes. In addition, the significant change of the size indicates that GagΔP6 undergoes some conformational changes when both ΨRNA and PI(4,5)P2 are present [38, 66]. Based on the sizes measured a schematic of the dimer and tetramer structure is proposed in Fig 11. From the size of the tetramers the spacing between the GagΔP6 molecules is around 7 nm close to the value of 8nm reported in studies using cET [18,19]. The substantial increases in the percentages of dimer and tetramer indicate that both ΨRNA and PI(4,5)P2 can bind with GagΔP6 and collectively facilitate HIV GagΔP6 assembly as reported in other studies [57, 63].

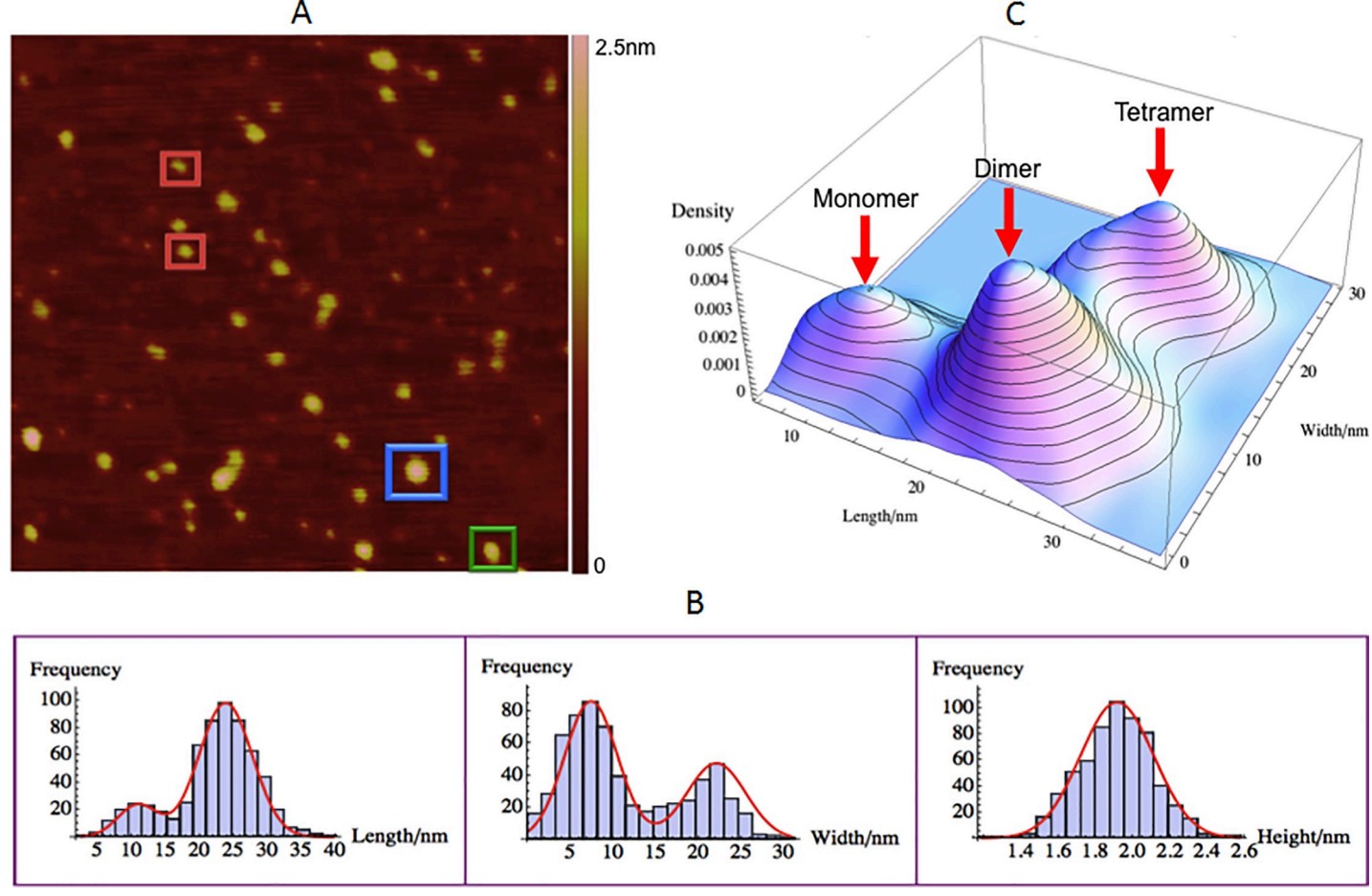

**Fig 10. PI(4,5)P2-ΨRNA-GagΔP6 interaction complex size and, morphology.** Mixture of PI(4,5)P2-ΨRNA-GagΔP6 (0.5μM : 0.5μM : 0.5μM) complex on negatively charged mica(-). (A) A typical AFM image with a scan size of 500nm×500nm. The height color bar scale is 2.5nm. A few characteristic PI(4,5)P2-ΨRNA-GagΔP6 complexes are boxed: monomer (red), dimer (green) and tetramer (blue). (B) Histogram for length (left), width (middle), and height (right). Shown in red are normal distribution fits to the peaks. (C) Three dimensional smooth histograms, where monomer, dimer, and tetramer are indicated by red arrows. The mean height is 1.91 ± 0.01nm for all three complexes. The first peak with a mean length of 10.9 ± 0.3nm and width 7.4 ± 0.1nm corresponds to the PI(4,5)P2-ΨRNA-GagΔP6 monomer. The second peak with a mean length of 23.8 ± 0.2nm and width 7.4 ± 0.1nm corresponds to the PI(4,5)P2-ΨRNA-GagΔP6 dimer. The third peak with a mean length of 23.8 ± 0.2nm and width 22.1 ± 0.2nm corresponds to the PI(4,5)P2-ΨRNA-GagΔP6 tetramer. The total number of samples was 616 and the experiment was repeated twice.

## Conclusion

The AFM technique was utilized to study the morphology of GagΔP6 (0.5μM), ΨRNA (0.5μM), and their binding complexes with the lipid PI(4,5)P2 in HEPES buffer with 0.1 Å vertical and 1nm lateral resolution. For the calibration 2nm diameter Au spheres were used. TAR-polyA RNA was used as a negative RNA control. The morphology of specific complexes GagΔP6-ΨRNA (0.5μM : 0.5μM), GagΔP6-TARpolyA RNA (0.5μM : 0.5μM), GagΔP6-PI(4,5)P2 (0.5μM : 0.5μM) and PI(4,5)P2-ΨRNA- GagΔP6 (0.5μM : 0.5μM : 0.5μM) were studied. They were imaged on either positively charged or negatively charged mica substrates depending on the net charges carried by the respective materials. The size and morphology of both ΨRNA and TARpolyA RNA measured was used to validate the technique in comparison with literature. For the ΨRNA, from the measured size two distinct populations corresponding to monomers and dimers were observed. In the case of TARpoly A RNA only the monomer population was found for the concentrations studied. The morphology of GagΔP6, being net

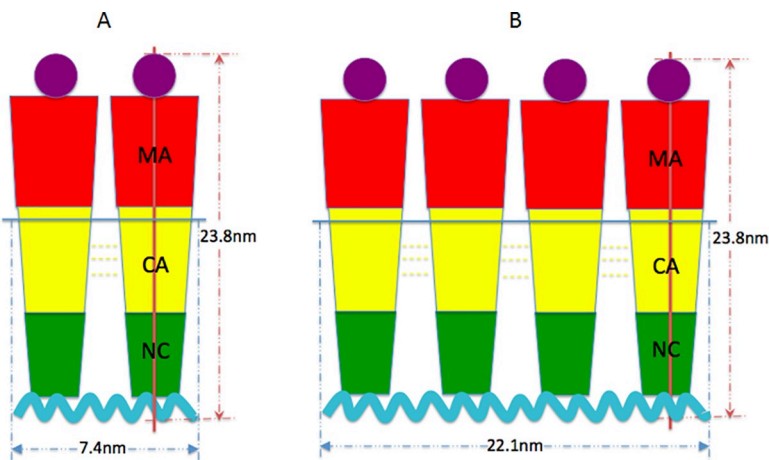

**Fig 11. Model of PI(4,5)P2-ΨRNA-GagΔP6 interaction complex.** Rough models of PI(4,5)P2-ΨRNA-GagΔP6 complexes based on the measured mean height and width. (a) Dimer complex, (b) Tetramer complex. MA domain is in red, CA domain is in yellow, NC domain is in green, ΨRNA is in cyan, and PI(4,5)P2 is in purple.

positively charged, was measured using the AFM on a freshly cleaved negatively charged mica (-) substrate in HEPES solution with a low salt concentration of 50mM NaCl. Three distinct size populations were found. They were found to correspond to 59% monomer, 35% dimer and 6% tetramer form of GagΔP6. The presence of the multimers was confirmed by gel electrophoresis. The addition of ΨRNA to 0.5μM GagΔP6 was observed to promote multimerization of the GagΔP6 leading to higher populations of dimers (14% increase) and tetramers (10% increase). The small change in size of the complexes confirmed the binding of the ΨRNA to GagΔP6. The addition TARPolyA RNA to GagΔP6 did not modify the GagΔP6 population distribution of monomers, dimers and tetramers. The interaction of PI(4,5)P2 with GagΔP6 complex was next measured on negatively charged mica(-). From the population distribution of the monomers (47%), dimers (41%) and tetramers (12%), it was concluded that PI(4,5)P2 promotes multimerization of GagΔP6 but not to the extent observed with ΨRNA.

The PI(4,5)P2-ΨRNA-GagΔP6 ternary complex was next studied using a negatively charged mica(-) surface. Both the population and size distribution of the GagΔP6 was completely different from that of the GagΔP6-ΨRNA and the GagΔP6- PI(4,5)P2 binary complexes. The population distribution of the monomers (17%), dimers (51%) and tetramers (32%) was significantly different. Given the large fraction of dimers and tetramers it was concluded that the presence of both PI(4,5)P2 and ΨRNA promotes extensive multimerization of GagΔP6. In addition, the significant change of size indicates that GagΔP6 undergoes a conformational change to a 23.8nm rod like shape when both ΨRNA and PI(4,5)P2 are present. From the size of the tetramers the spacing between the GagΔP6 molecules is around 7 nm which is consistent with studies using cET in Ref [18, 19]. The substantial increases in the percentages of dimer and tetramer indicate that both ΨRNA and PI(4,5)P2 can bind with GagΔP6 and collectively facilitate HIV GagΔP6 assembly as reported using other techniques.

The gel electrophoresis data presented in the supplementary section for the GagΔP6, ΨRNA and PI(4,5)P2 lipid combinations provide complementary confirmation for the different complexes that are present in the same solution mixtures analyzed with the AFM. The size and population distribution measured for the various complexes are distinct and show changes for the complexes on the addition the ΨRNA and PI(4,5)P2 to the GagΔP6 solution.

## Supporting information

**S1 File. Supporting information.**
(DOCX)

## Acknowledgments

We would like to acknowledge discussions with Karin Musier-Forsyth, Erick D. Olson and Ioulia Rouzina. We would like to thank Erick Olson and Karin Musier-Forsyth for providing the in vitro transcribed RNAs. We also acknowledge discussions with Alan Rein and thank him and S.A.K. Datta for providing the GagΔP6 molecules.

## Author Contributions

**Conceptualization:** Shaolong Chen, Jun Xu, Roya Zandi, Sarjeet S. Gill, Umar Mohideen.

**Data curation:** Shaolong Chen.

**Formal analysis:** Shaolong Chen, Jun Xu.

**Funding acquisition:** Umar Mohideen.

**Investigation:** Shaolong Chen, Jun Xu, Roya Zandi, Umar Mohideen.

**Methodology:** Shaolong Chen, Jun Xu, A. L. N. Rao, Sarjeet S. Gill, Umar Mohideen.

**Project administration:** Shaolong Chen, Umar Mohideen.

**Resources:** A. L. N. Rao, Sarjeet S. Gill, Umar Mohideen.

**Software:** Shaolong Chen, Mingyue Liu.

**Supervision:** Umar Mohideen.

**Validation:** Shaolong Chen, A. L. N. Rao, Umar Mohideen.

**Visualization:** Shaolong Chen, Mingyue Liu.

**Writing – original draft:** Shaolong Chen, Umar Mohideen.

**Writing – review & editing:** A. L. N. Rao, Roya Zandi, Umar Mohideen.

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
