## [Decision Letter · Decision Letter 0]

12 Nov 2019

PONE-D-19-27871

Investigation of HIV-1 Gag binding with RNAs and Lipids using Atomic Force Microscopy

PLOS ONE

Dear Dr. Chen,

Thank you for submitting your manuscript to PLOS ONE. After careful consideration, we feel that it has merit but does not fully meet PLOS ONE’s publication criteria as it currently stands. Therefore, we invite you to submit a revised version of the manuscript that addresses the points raised during the review process.

The manuscript was reviewed by two experts in the field and both raised some concerns that need to be addressed before the manuscript is considered for publication.   

We would appreciate receiving your revised manuscript by Dec 27 2019 11:59PM. To enhance the reproducibility of your results, we recommend that if applicable you deposit your laboratory protocols in protocols.io, where a protocol can be assigned its own identifier (DOI) such that it can be cited independently in the future. For instructions see: http://journals.plos.org/plosone/s/submission-guidelines#loc-laboratory-protocols

We look forward to receiving your revised manuscript.

Kind regards,

Jamil S Saad, Ph.D

Academic Editor

PLOS ONE

Journal Requirements:

Reviewers' comments:

Reviewer's Responses to Questions

**Comments to the Author**

1. Is the manuscript technically sound, and do the data support the conclusions?

Reviewer #1: Partly

Reviewer #2: Yes

2. Has the statistical analysis been performed appropriately and rigorously? 

Reviewer #1: Yes

Reviewer #2: Yes

3. Have the authors made all data underlying the findings in their manuscript fully available?

Reviewer #1: Yes

Reviewer #2: Yes

4. Is the manuscript presented in an intelligible fashion and written in standard English?

Reviewer #1: No

Reviewer #2: Yes

5. Review Comments to the Author

Reviewer #1: The manuscript entitled “Investigation of HIV-1 Gag binding with RNAs and Lipids using Atomic Force Microscopy” by Chen et al, focused to investigate the molecular structure of complexes involving Gag protein with its multiple targets including �-RNA, TAR polyA RNA and lipids PI(4,5)P2 by using Atomic force microscopy which provides direct visual information at the single molecule level. The size of individual components was also visualized and the presence of dimers, tetramers were observed in some cases. Although, the present study show some interesting indication of multimerization of Gag�P6 in presence of both PI(4,5)P2 and �-RNA, the following concerns should be clarified before publication:

1. The major concern in the whole study is how the authors identify the complexes in their AFM images (Fig 7-10)? The images of RNA have distinct linear shape. As soon as the protein is added the whole complex become spherical or globular. At this stage, it becomes really difficult to assign whether the complex is really formed or not. The measurement of height, length, width is not enough to characterize the complex formation. This drawback poses concern on the conclusion drawn from the data set shown in Fig 7-10.

To avoid this ambiguation, authors could make a construct where their specific RNA is attached at the end of double stranded DNA (~300 bp). In this construct, the advantage would be whenever the protein-RNA complex would be formed, it would show up as a feature containing a blob at the end of the dsDNA, where the RNA construct is situated. It would unambiguously show that the complex is formed, and further similar analysis could be done on those features.

Another possibility to avoid this concern is to perform AFM time-lapse imaging, where the RNA would be imaged first and then different proteins, lipids and their combination could be added to visualize the complex formation in situ. Unless these issues are clarified, it is difficult to comment on the conclusions.

2. Authors should elaborate on why the width of the �-RNA becomes double in the dimer. As the length of the RNA is increased the width value should remain the same.

3. Authors mentioned that the Gag�P6 is cylindrical in shape, whereas in AFM image they appeared spherical. The reason behind that should be clarified. They have provided the model in Fig. 6, but enough experimental evidence is missing.

4. How did the authors calculate the width for elongated complexes? Did they measure at three different regions and then average it?

5. Did the authors separate the excess lipids from the complexes? If free lipid were present lipid patches should be observed on the surface in Fig. 9.

6. Authors should compress the introduction section. The elaborate details of the Gag structure is not necessary rather the rationale of the present work should be mentioned clearly.

Reviewer #2: This is the review of the manuscript entitled “Investigation of HIV-1 Gag binding with RNAs and Lipids

using Atomic Force Microscopy” by Chen et al. In this manuscript the authors used solution AFM experiments to probe the multimerization and conformations of Gag in the presence of PIP2 and RNA. The authors show good resolution of the AFM using 2nm gold particles and go on to resolve ΨRNA as well as the TARpolyA RNA molecules absorbed to mica surfaces. They show a significant fraction of dimer ΨRNA absorbed on the mica and detect the U shaped conformation of Gag. I have found the study to be sufficiently rigorous and well presented, however there are important shortfalls that need to be addressed before the paper is accepted for publication.

Major issues:

1- It is not clear how quantification of dimer Ψ RNA was performed? Based on the definition of the width measurements presented in the supplementary materials, the dimers should have much larger widths than 4 nm.

2- When GagΔP6 is observed on mica, the histograms show Gag molecules with a diameter of ~6 nm and length of 10 nm, the dimers show diameter of 6nm and length of 20 nm. Analyzing the AFM image however, It didn’t look like the identified dimers are longer, instead they were showing a larger overall diameter. This was also true for the tetramers, the authors should explain this discrepancy

3- The authors are assuming that what they observed is a stoichiometric representation of the ingredients, however their method only shows topological features with no specificity. To be more specific, when the authors say they show PIP2-GagΔP6-RNA complexes, there is no independent verification that the blub they are looking at actually contains all those components. If the authors can provide independent verification of complexes it will significantly strengthen the study, otherwise the authors should rewrite and de-emphasize the stoichiometry of their complexes. In short, the way the paper now reads, it implies independent verification, where there is none.

4- Basic mutagenesis of Gag would have helped the study become a lot more interesting, for example the G2A mutant?

5- There needs to be a significant discussion about what the mica surfaces are doing in to skew the selection of molecules from solution, also Gag molecules typically interact with lipid bilayers, how is that effecting the experimental results has to be addressed.

6. PLOS authors have the option to publish the peer review history of their article (what does this mean?). If published, this will include your full peer review and any attached files.

Reviewer #1: No

Reviewer #2: No

---

## [Author Response · Author response to Decision Letter 0]

24 Dec 2019

Dear Editor: 

Thank you for sending us the review reports and your positive feedback on the manuscript. We thank the reviewers for the careful reading and analysis of the manuscript. We agree with the reviewer #1 that size change alone is not sufficient to confirm binding. However, the goal in this manuscript was to study the change in the monomer/dimer/tetramer population statistics and structure of HIV Gag with the addition of RNA and the lipid PIP2, not just whether RNA binding alone has happened. The binding is itself is checked through gel electrophoresis. Following the suggestion of reviewer #1 we have compressed the introduction. We appreciate reviewer #2’s conclusion that the manuscript is “sufficiently rigorous and well presented.” We have provided answers to all the questions of both reviewers and added clarifications in the manuscript where necessary. The added sentences are highlighted in yellow in the manuscript file titled “Revised Manuscript With Track Changes” for the reviewers benefit. The complete response to the reviewers is uploaded. The requested original gel data is also uploaded as supporting information. 

Sincerely 

Umar Mohideen 

For all Authors

---

## [Decision Letter · Decision Letter 1]

7 Jan 2020

Investigation of HIV-1 Gag binding with RNAs and Lipids using Atomic Force Microscopy

PONE-D-19-27871R1

Dear Dr. Mohideen,

We are pleased to inform you that your manuscript has been judged scientifically suitable for publication and will be formally accepted for publication once it complies with all outstanding technical requirements.

With kind regards,

Jamil S Saad, Ph.D

Academic Editor

PLOS ONE

Additional Editor Comments (optional):

Reviewers' comments:

Reviewer's Responses to Questions

**Comments to the Author**

1. If the authors have adequately addressed your comments raised in a previous round of review and you feel that this manuscript is now acceptable for publication, you may indicate that here to bypass the “Comments to the Author” section, enter your conflict of interest statement in the “Confidential to Editor” section, and submit your "Accept" recommendation.

Reviewer #1: All comments have been addressed

Reviewer #2: All comments have been addressed

2. Is the manuscript technically sound, and do the data support the conclusions?

Reviewer #1: Partly

Reviewer #2: Yes

3. Has the statistical analysis been performed appropriately and rigorously? 

Reviewer #1: Yes

Reviewer #2: Yes

4. Have the authors made all data underlying the findings in their manuscript fully available?

Reviewer #1: Yes

Reviewer #2: Yes

5. Is the manuscript presented in an intelligible fashion and written in standard English?

Reviewer #1: Yes

Reviewer #2: Yes

6. Review Comments to the Author

Reviewer #1: (No Response)

Reviewer #2: My concerns have been sufficiently addressed with changes provided in this revision. I recommend publication of this article.

7. PLOS authors have the option to publish the peer review history of their article (what does this mean?). If published, this will include your full peer review and any attached files.

Reviewer #1: Yes: Siddhartha Banerjee

Reviewer #2: No

---

## [Editor Report · Acceptance letter]

16 Jan 2020

PONE-D-19-27871R1 

Investigation of HIV-1 Gag binding with RNAs and Lipids using Atomic Force Microscopy 

Dear Dr. Mohideen:

I am pleased to inform you that your manuscript has been deemed suitable for publication in PLOS ONE. Congratulations! Your manuscript is now with our production department. 

With kind regards,

on behalf of

Dr. Jamil S Saad 

Academic Editor

PLOS ONE